# Natural Products of Marine Origin for the Treatment of Colorectal and Pancreatic Cancers: Mechanisms and Potential

**DOI:** 10.3390/ijms23148048

**Published:** 2022-07-21

**Authors:** Nasrin Fares Amer, Tal Luzzatto Knaan

**Affiliations:** Department of Marine Biology, The Leon H. Charney School of Marine Sciences, University of Haifa, 199 Aba Koushy Ave., Mount Carmel, Haifa 3498838, Israel; nfares@campus.haifa.ac.il

**Keywords:** marine natural products, colon cancer, pancreatic cancer, apoptosis

## Abstract

Gastrointestinal cancer refers to malignancy of the accessory organs of digestion, and it includes colorectal cancer (CRC) and pancreatic cancer (PC). Worldwide, CRC is the second most common cancer among women and the third most common among men. PC has a poor prognosis and high mortality, with 5-year relative survival of approximately 11.5%. Conventional chemotherapy treatments for these cancers are limited due to severe side effects and the development of drug resistance. Therefore, there is an urgent need to develop new and safe drugs for effective treatment of PC and CRC. Historically, natural sources—plants in particular—have played a dominant role in traditional medicine used to treat a wide spectrum of diseases. In recent decades, marine natural products (MNPs) have shown great potential as drugs, but drug leads for treating various types of cancer, including CRC and PC, are scarce. To date, marine-based drugs have been used against leukemia, metastatic breast cancer, soft tissue sarcoma, and ovarian cancer. In this review, we summarized existing studies describing MNPs that were found to have an effect on CRC and PC, and we discussed the potential mechanisms of action of MNPs as well as future prospects for their use in treating these cancers.

## 1. Introduction

Worldwide, an estimated 19.3 million new cancer cases and about 10.0 million cancer deaths occurred in 2020. Female breast cancer is the most commonly diagnosed cancer, with an estimated 2.3 million (~11.7%) new cases every year, followed by lung (~11.4%), colorectal (~10.0%), prostate (~7.3%), stomach (~5.6%), and pancreatic (~2.5%) cancers. Lung cancer is the leading cause of cancer death (~18%), followed by colorectal (~9.4%), liver (~8.3%), stomach (~7.7%), female breast (~6.9%), and pancreatic (~4.5%) cancers [1,2].

Gastrointestinal cancer refers to malignant of accessory organs of digestion, and it includes colorectal cancer (CRC) and pancreatic cancer (PC). In 2020, 1.9 million new CRC cases and 0.9 million CRC deaths were estimated worldwide [1]. The CRC incidence is higher in developed countries; however, the number of cases is increasing in non-developed countries every year [3]. PC is a malignant tumor that usually occurs as a pancreatic adenocarcinoma. It has a poor prognosis and high mortality, with an estimated 5-year relative survival of 11.5% [4]. According to the Global Cancer Observatory, there were 495,773 estimated new PC cases and 466,003 deaths in 2020 [1]. In 2018, PC was estimated to be the eleventh most common cancer in the world, with 458,918 new cases and 432,242 deaths (4.5% of all deaths caused by cancer). According to epidemiological studies, PC incidence is increasing every year regardless of gender [5,6].

Cancer care usually requires teamwork by doctors in multiple disciplines who combine different types of treatments. Treatment options depend on several factors, including the type and stage of cancer, possible side effects, and overall health. The treatment options for CRC and PC are surgery, radiation therapy, chemotherapy, targeted therapy, and immunotherapy [7]. Only about 20% of people diagnosed with PC are able to access surgery due to late stage diagnosis, at which point the disease has already spread. External beam radiation therapy is the type of therapy used most often to treat PC [8]. Medications are also given individually or as a cocktail as part of the treatment plan [8]. Drugs used in chemotherapy induce severe side effects that greatly affect the life quality of patients, including weakness, hair loss, nausea, vomiting, diarrhea, abdominal cramps, mouth sores, dry mouth, and numbness [9]. Therefore, there is an urgent need to identify new chemical targets and active compounds with higher selective potency or new cellular targets.

Natural products, which are chemical compounds produced by organisms, have been used for centuries as remedies for various illnesses. Since ancient times, humans have utilized natural products from various sources, such as plants, marine organisms, and microorganisms, for diverse applications. This history continues to provide inspiration for alternative agents to fight various diseases, but sources from the marine environment remain relatively underexplored [10].

## 2. Marine Natural Products (MNPs) with the Potential to Treat Cancer

MNPs are compounds isolated from marine microorganisms and phytoplankton, algae, sponges, mollusks, tunicates, echinoderms, mangroves, and others. Since the 1970s, marine organisms have played an important role in the discovery of novel biologically active compounds, and MNPs have become a source of bioactive secondary metabolites with the potential to treat diseases, including cancer [11,12,13].

The screening of 3019 compounds from the MNPs library identified that four compounds have potential therapeutic anticancer activity through the mammalian target of the rapamycin pathway (mTOR). mTOR regulates different cellular processes including cell growth and cell proliferation [14]. In addition, marine compounds such as glycosides, alkaloids, saponins, lipids, terpenes, ribose, steroids, xanthones, ethers, lignins, coumarins, carbazoles, azaphilones, nucleosides, polyketides, and quinones have been shown to have high cytotoxic activities against 121 mammalian cancer cell lines including breast, colon, melanoma, lung and pancreatic human tumor cells [15]. Studies have also shown that nucleotides, proteins, peptides and amides compounds isolated from MNPs have anti-angiogenic, anti-proliferative and anti-metastasis activities, and lead to cell cycle arrest, and the induction of apoptosis [16,17]. MNPs compounds, such as polysaccharides, polyphenols, carotenoids, and terpenoids, that were isolated from seaweeds have been shown to prevent cancer growth inducing programing cell death and cell cycle arrest [18]. Soft corals-derived sinularin and dihydrosinularin also show anti-cancer activity [19].

Furthermore, recent studies investigated scalarine, a compound isolated from different marine sponges, and demonstrated that it can reduce the level of the receptor for advanced glycation end products (RAGE) and inhibit autophagy in human pancreatic cell lines (PANC-1 and MIA PaCa-2). RAGE has lately become a chemotherapeutic target for both treatment and chemoprevention. RAGE seems to be a key regulator of the inflammatory, stress, and survival pathways that contribute to PC carcinogenesis, chemotherapy resistance, increased proliferation, and a high risk of metastasis. It has been shown that PC tumors express RAGE, but not the surrounding epithelial tissues [20].

The first marine compound used in clinical trials as an anti-cancer drug was Didemnin B, which was extracted from the ascidian *Trididemnum solidum* [21,22]. Didemnin B targets elongation factor 1α, which is responsible for the enzymatic delivery of aminoacyl tRNAs to the ribosome. It plays an important role in protein translation and participates in several processes required for cell growth and proliferation. However, this compound failed in clinical trials due to its toxicity [23]. Another compound, Spongistatin 1, was isolated from the marine sponges *Spirastrella spinispirulifera* and *Hyrtios erecta*. It was found to potently induce cell death in primary acute leukemic cells and prevent the long-term survival of leukemic cell lines. Spongistatin 1 induced apoptosis through the caspase-dependent pathway by degrading the X-linked inhibitor of apoptosis protein (IAP), releasing cytochrome c from the mitochondria, and activating caspase-9 [24]. However, Spongistatin 1 has still not been approved as a drug.

Another MNP is the marine actinomycete-derived cytotoxic secondary metabolite Salinosporamide A, which is a potent proteasome inhibitor [25]. This proteasome was effectively validated as a target in cancer chemotherapy, and it was approved for the treatment of multiple myeloma in 2003 [26,27]. An evaluation of this compound’s mechanism of action indicated that it inhibits the activation of the nuclear factor kappa-light-chain-enhancer of activated B cells (NFκB) and 13 genes regulated by NFκB, a hallmark downstream event of proteasome inhibition that is involved in inducing apoptosis [25].

To date, MNPs from different marine species of sponges, tunicates, mollusks, and cyanobacteria have been approved as anti-cancer marine molecules, including Cytosar-U^®^, Depocyt^®^, Halaven^®^, Fludara^®^, Arranon^®^, Yondelis^®^, Adcetris^®^, Polivy^®^, Farydak^®^, Aplidine^®^, Zepzelca™, Blenrep™, Aidixi™, TIVDAK™, and PADCEV™ [28,29,30]. These drugs have several side effects including severe ones such as a fast or slow heart rate, difficulty breathing, high blood sugar (hyperglycemia), peripheral neuropathy, neutropenia, leukopenia, hypoesthesia, increased conjugated blood bilirubin, decrease in platelets (thrombocytopenia), low blood cell counts, infusion related reactions, and muscle pain (myalgia). In addition, several MNPs are currently being evaluated in different phases of human clinical trials [31,32]. In addition, recent in vitro, in vivo, and clinical studies have shown several marine compounds such as alkaloids, peptides, terpenoids, poly saccharides, and carotenoids have antitumor effects on CRC [33]. However, many MNPs are toxic and their use was terminated in clinical studies. Therefore, continued efforts in the field of marine drug discovery are expected to reveal more potent bioactive compounds with diverse mechanisms of action.

## 3. Mechanisms of Action of MNPs in CRC and PC

Numerous MNPs have been found to affect CRC and PC cells via different mechanisms (Table 1 and Table 2). The main mechanisms are caspase activation pathways, the inhibition of anti-apoptotic factors, tubulin interaction, cell cycle arrest, activation of the p53 pathway and increased intracellular reactive oxygen species (ROS) accumulation (Table 3).

### 3.1. Induction of Apoptosis through Caspase Activation

Some MNPs were found to affect CRC and PC cells through caspase activation pathways. Caspases are protease enzymes that have essential roles in programmed cell death (apoptosis) [62]. Caspases-2, -3, -6, -7, -8, -9, and -10 are activated during apoptosis. Caspases-2, -8, -9, and -10 are called apical or upstream caspases, and are responsible for initiating caspase activation cascades. Caspases-3, -6, and -7 are called downstream caspases, and are responsible for the actual destruction of the cell during apoptosis [63]. During activation, apical caspases distribute death signals and activate downstream caspases in a cascade-like manner [64]. Downstream caspases then cause direct cellular structures disintegration, cellular metabolism disruption, cell death, inhibitory proteins inactivation, and additional destructive enzymes activation [65]. For example, marine sponge metabolites ilimaquinone and ethylsmenoquinone were found to activate caspase-3 and induce apoptosis via this pathway [40].

Many apoptosis-inducing stimuli activate caspases through one of the following three major pathways: (i) the mitochondrial pathway (intrinsic pathway), (ii) the death receptor pathway (extrinsic), and (iii) the cytotoxic T lymphocytes/natural killer cells (CTL/NK)-dependent pathway [63] (Figure 1).

#### 3.1.1. Mitochondrial Apoptosome-Driven Pathway of Caspase Activation (Intrinsic)

Mitochondrial outer membrane permeability is a key mechanism of caspase activation triggered by stimuli of cytotoxic drugs and other extracellular pressures [63]. These stimuli trigger the release of mitochondrial components such as cytochrome c. Cytochrome c assembles with apoptosis protease-activating factor-1 (Apaf-1), deoxyadenosine triphosphate, and procaspase-9 to form a heptameric rota-shaped caspase-activating complex called the apoptosome, leading to caspase-9 activation [66]. Pro-apoptotic and anti-apoptotic members of the B-cell lymphoma 2 (Bcl-2) protein family control outer mitochondrial membrane (OMM) permeability and regulate the release of cytochrome c and other components of the mitochondrial intermembrane space [67]. Apoptosis-inhibitory members of the Bcl family, such as Bcl-2 and Bcl-extra-large (Bcl-xL), inhibit the release of mitochondrial factors, resulting in the inhibition of apoptosis. In contrast, pro-apoptotic members, such as Bcl-2-associated X protein (Bax), Bcl-2-antagonist/killer (Bak), Bcl-2 homology (BH)-3 proteins, BH3-interacting domain death agonist (Bid), Bcl-2 associated agonist of cell death (Bad), and Bcl-2 interacting mediator of cell death (Bim), promote mitochondrial factor release and induce apoptosis (Figure 1A).

The precise molecular mechanism by which Bcl-2 family proteins regulate OMM permeability is controversial [68]. Upon the activation of the apoptosome, Caspa-se-9 can transmit a death signal by activating downstream caspases [63]. A cascade of caspase activation events is initiated by the apoptosome [64]. In this cascade, caspase-3 and caspase-7 appear to be simultaneously activated by caspase-9 in the apoptosome. Activated caspase-3 then activates caspase-2 and caspase-6, leading to apoptosis [63].

Ryu et al. found that an extract prepared from the green seaweed *Ulva fasciata* activates the mitochondria-dependent apoptotic pathway by downregulating Bcl-2 and upregulating Bax, resulting in the release of the apoptosis-inducing factor cytochrome c from the mitochondria into the cytosol. Subsequently, caspase-9 and caspase-3 are activated and induce apoptosis [44].

#### 3.1.2. Death Receptor Pathway of Caspase Activation (Extrinsic)

The extrinsic cell death pathway is mediated by activating the tumor necrosis factor (TNF) family of death receptors. This family includes TNF receptor 1 (TNF-R1), CD95 (APO-1, Fas), TNF-related apoptosis-inducing ligand receptors (TRAIL-R1 and TRAIL-R2, also known as DR4 and DR5), and DR3 and DR6. Extrinsic cell death is initiated by the recruitment of adaptor proteins such as the FAS-associated death domain protein (FADD) that binds to the death-effector domain containing procaspases to generate the death-inducing signaling complex (DISC), resulting in the activation of caspase-8. Caspase-8 directly cleaves and activates caspase-3, the executioner enzyme of apoptosis [62] (Figure 1B).

The extrinsic pathway of apoptosis can be linked to the intrinsic pathway by cleavage of Bid. Bid activates Bak and Bax in the mitochondria, leading to the release of cytochrome c. In the cytoplasm, cytochrome c binds to the adaptor molecule Apaf-1, which interacts with procaspase-9 to form the apoptosome. The activation of Caspase-9, leads to the activation of the downstream caspases-3, -6, and -7, which induce apoptosis. IAP directly binds to caspases and inhibits their enzymatic activity. IAP inhibition is countered by the second mitochondria-derived activator of caspases (SMAC) [62]. Kim et al. reported that fucoidan extract from the marine brown alga *Turbinaria conoides* significantly increased the levels of death receptors FAS and TRAIL in HT-29 human CRC cells via this pathway [45].

#### 3.1.3. CTL/NK-Dependent Pathway

CTLs and NK cells induce apoptosis in tumor cell targets by various means. An important pathway utilized by these cells involves the release of cytolytic granules containing various enzymes capable of inducing apoptosis in their target cells [69]. These granules contain proteins such as granzyme B, which directly activate caspase-3 and caspase-8 and lead to the cleavage of Bid [70,71]. Therefore, CTL-mediated rapid apoptosis requires granzyme B [63] (Figure 1C).

In cancer treatment, one of the goals is to develop therapies that selectively induce apoptosis without affecting normal tissues. The induction of caspase activation or blocking caspase inhibitors are promising strategies for cancer treatment [72]. Joung et al. reported that sulfated polysaccharide (p-KG03) produced by marine red tide microalga *Gyrodinium impudicum* strain KG03 possesses immunostimulatory effects and enhanced tumoricidal activity of macrophages and NK cells in vivo in mouse lymphoma cells [73]. However, to date, there have been no reports of MNPs activating this pathway in CRC and PC.

### 3.2. Inhibition of Anti-Apoptotic Factors

One mechanism of resistance to apoptotic stimuli in cancer cells involves the overexpression of the IAP family of proteins. IAPs include X-linked IAP (XIAP), cellular IAP-1 (cIAP-1), and cIAP-2 [46,74]. These proteins have been shown to prevent apoptosis by binding to caspases, including initiators and effectors, and thereby protecting them from cleavage and activation [59,74].

Kim et al. reported that the fucoidan extract from brown algae attenuated the levels of XIAP and survivin, which are members of the IAP family [45]. Additionally, Zhang et al. found that libertellenone-H (LH) isolated from Arctic marine fungi inhibits the thioredoxin (TRX) system, which leads to cellular stress and cell death in human PC cell lines [61]. The TRX system is expressed in all living cells and has a variety of biological functions related to cell proliferation and apoptosis. It also is a key part of the antioxidant system that defends against oxidative stress. Moreover, the TRX system plays critical roles in the immune response, virus infection, and cell death by interacting with the thioredoxin interacting protein [75]. The TRX system also helps tumor cells evade apoptosis by directly binding to the apoptosis signal regulating kinase and the tumor suppressor gene phosphatase and tensin homolog (PTEN) [76,77].

### 3.3. Interaction of MNPs with Tubulin to Cause Anti-Mitotic Activity

Microtubules are elements of the cytoskeleton and play important roles in many cellular functions, including intracellular transport, motility, morphogenesis, and cell division [78]. Microtubules are heterodimers composed of α-β subunits that assemble to form the protofilaments of the tube [79]. The polymerization and depolymerization of microtubules are critical processes for cell division machinery. Checchi et al. showed that disrupting these processes with microtubule binding drugs could be a useful tool to inhibit cell proliferation [80].

Leiodermatolide, a MNP isolated from a deep-water sponge, has a cytotoxic effect on human PC and CRC cells. It induces cell cycle arrest and apoptosis, and it affects the microtubules required for spindle formation and chromosome segregation [57]. Another compound, PM060184, isolated from the marine sponge *Lithoplocamia lithistoides* is considered to be a tubulin binding agent with potent anti-tumor activity [38].

### 3.4. Suppression of Cell Cycle Progression

The cell cycle is a four-stage process in which the cell increases in size (G1 stage), copies its DNA (S stage), prepares to divide (G2 stage), and undergoes mitosis (M stage). The cell cycle is controlled at three checkpoints. The integrity of the DNA is assessed at the G1, G2, and M checkpoints, and the cell cycle is regulated by a family of enzymes called the cyclin-dependent kinases (CDKs) [81].

Tsai et al. studied the mechanisms underlying G2/M cell cycle arrest induced by 5-*epi*-sinuleptolide, a compound isolated from the marine soft coral *Sinularia leptoclados* [54]. They measured the expression levels of several G2/M progression-related proteins and found that 5-*epi*-sinuleptolide increased the expression of cyclin B1 and affected the phosphorylation status of CDK1 in a dose-dependent manner. Sustaining high cyclin B1–CDK1 activity resulted in cell arrest in the M phase and caused cell cycle arrest. 5-*epi*-sinuleptolide also suppressed the expression of cyclin D, which is an important cell cycle regulator throughout the cell cycle [82]. Ono et al. reported that p53 induced G1/S phase or G2/M arrest via p21 inhibition of CDKs [83]. Their results indicated that 5-*epi*-sinuleptolide induced p21 expression but not p53 expression. Therefore, Tsai et al. suggested that 5-*epi*-sinuleptolide mediates cell cycle arrest through a p53 independent pathway [82].

Many other MNPs have been reported to suppress cell cycle progression [36,40,44,48,51,56,57]. For example, Bae et al. reported that Asperphenin A, a lipopeptidyl benzophenone metabolite isolated from the large-scale cultivation of marine-derived *Aspergillus* sp. fungus, inhibits the growth of cells through G2/M cell cycle arrest followed by apoptosis in RKO human CRC cells [48].

### 3.5. The Role of NFκB and p53 in Apoptosis

NFκB is involved in the development of cancer. Wang et al. reported that it is constitutively activated in PC cells but not in normal cells [84]. Nakanishi and Toi found that the constitutive activation of NFκB induces anti-apoptotic genes and inhibits apoptosis [85]. The cytoplasmic factor IkappaB (IκB) binds to NFkB and inhibits its translocation to the nucleus. After translocation to the nucleus, NFkB binds to a specific promotor and regulates the transcription of many genes associated with oncogenesis and tumor progression. Several studies have shown that the inhibition of NFκB results in the induction of apoptosis in cancer cells [86,87,88]. Therefore, the inhibition of NFκB antagonizes the survival of cancer cells and induces apoptosis [85].

While NFκB is known to inhibit apoptosis, the transcription factor p53, a tumor suppressor gene, is an inducer of apoptosis. p53 plays a regulation or progression role in the cell cycle, apoptosis, and genomic stability through multiple mechanisms, including cell cycle arrest, the activation of DNA repair proteins, and induction of apoptosis [60]. p53 limits cell proliferation by the induction of transient G1 cell cycle arrest or apoptosis. The molecular explanation for the p53 mediated growth arrest response is based on its ability to act as a sequence-specific DNA binding transcription factor. p53 has several downstream target genes, such as p21, Mouse Double Minute 2 homolog (MDM2), Growth Arrest and DNA Damage Inducible Alpha (GADD45), cyclin G, and Bax, whose expression products act as regulators of various aspects of cell growth [89,90]. The activation of p53 negatively regulates the expression and NFκB activity of the RelA (p65) subunit [60].

Lee et al. found that the marine sponge metabolites ilimaquinone and ethylsmenoquinone induce p53 expression, which leads to cell cycle arrest at the G2/M phase and the induction of apoptosis [40]. In addition, sugars from red seaweed induce p53 expression and the induction of apoptosis [60]. The fucoidan extract from marine brown alga increases the level of death receptors and decreases the level of anti-apoptotic factors in CRC cells [45]; in PC cells, the extract targets p53–NFκB crosstalk and induces apoptosis [60]. Several studies have shown that fucoidan has an anti-tumor effect, including inhibiting growth, metastasis and angiogenesis and inducting the apoptosis of various cancer tumor cells in vitro and in vivo [45,60,91,92,93,94]. Most in vitro experiments have demonstrated that fucoidan at the cytotoxic concentration for tumor cell lines has no effect on normal cell growth and mitosis [95,96]. In an in vivo experiment, Li et al. found that the administration of 300 mg/kg of fucoidan to rats by oral gavage daily for 6 months had no significant adverse effects. However, doses of 900–2500 mg/kg caused coagulopathy, and the clotting time was significantly prolonged [97]. Alekseyenko et al. transplanted Lewis lung adenocarcinoma into C57 mice and found that when mice were repeatedly injected with 10 mg/kg of fucoidan, the drug showed significant anti-tumor (33% tumor growth inhibition) and anti-metastatic activity (29% reduction) [98]. Etman et al. reported similar results for PC [99]. Han et al. reported that fucoidan induces the suppression of human colon cancer cell (HT-29) proliferation and possesses anti-cancer effects via the protein kinase B pathway (AKT). This pathway induces the G1 phase-associated upregulation of p21 expression and suppresses the growth of HT-29 CRC cells [100].

### 3.6. Increased Intracellular ROS Accumulation and Induction of Apoptosis

Oxygen molecules are a diradicals and are not reactive compared to other molecules. However, incomplete oxygen reduction leads to the formation of more chemically reactive oxygen species (ROS), which include superoxide anion (O_2_^−^), hydrogen peroxide (H_2_O_2_), and hydroxyl radical (radical OH). Due to their strong chemical reactivity, ROS have traditionally been thought to mediate only oxygen toxicity. ROS have been implicated in oxidative stress mediated pathology, as they are considered disruptive agents that can structurally and functionally affect macromolecules such as nucleic acids, proteins, and lipids [101]. Several studies found that the accumulation of ROS decreases mitochondrial membrane potential (MMP) and mitochondria depolarization. The presence of ROS leads to cell cycle arrest at the G2/M phase, followed by the accumulation of DNA damage and induction of apoptosis [102,103]. Other researchers reported that marine extracts from nudibranchs, brown algae, and Arctic fungi trigger the intracellular accumulation of ROS in cancer cells originated from human CRC and PC, leading to apoptosis [36,58,61]. LH, a pimarane diterpenoid isolated from the Arctic marine fungus *Eutypella* sp. D-1, has effective cytotoxicity against a range of cancer cells and induces ROS accumulation. Further, Zhang et al. showed that LH exerts anti-proliferative activity against four PC cell lines in a dose-dependent manner, with IC50 values of 3.21, 0.67, 2.78, and 5.53 µM in PANC-1, SW1990, AsPC-1, and BxPC-3, respectively, after 48 h of treatment. In contrast, the IC50 value of LH for a normal pancreatic cell lines (HPDE6-C7) is 10.86 µM [61]. The clear difference between the IC50 value of LH between cancer cells and normal cells highlights LH as a potential drug lead.

## 4. Conclusions

MNPs have great potential as new compounds that can assist in the prevention and treatment of cancer, but extensive exploration is needed. Over the past 50 years, many MNPs with beneficial effects on the prevention and treatment of various types of cancer have been reported. For example, cytarabine, eribulin mesylate, brentuximab vedotin, and trabectidine are marine-based drugs used against leukemia, metastatic breast cancer, soft tissue sarcoma, and ovarian cancer [104,105]. MNPs compounds have different activities including the inhibition of the transformation of normal cells into tumor cells, halting tumor cell growth and microtumors development, and inducing apoptosis. A higher consumption of sea food is suggested as a promising strategy to prevent cancer [106,107]. Many marine edible organisms contain lipids enriched by polyunsaturated fatty acids (PUFAs), such as ω-3 fatty acids that have been shown in many experimental studies to suppress most forms of tumor development, including breast, colon, prostate, liver, and pancreatic tumors [41,108,109].

The National Cancer Institute provides researchers with the resources needed to better elucidate the role of food and nutrients in cancer prevention, but very little information is available about cancer prevention using MNPs [110]. To date, 13 MNPs have been tested in different phases of clinical trials, thus highlighting the potential of this source [111]. However, very few studies have tested the effect of MNPs on PC cells, and none are in the clinical stages for PC treatment. PC is one of the deadliest cancers and has only a few lifesaving treatment options. Therefore, additional research is necessary to develop new strategies for treatment with better effectiveness and fewer side effects.

Several MNPs affect CRC and PC cells via different mechanisms (Table 3), and the main ones are caspase activation pathways, the accumulation and inhibition of anti-apoptotic factors, tubulin interaction, cell cycle arrest, activation of the p53 pathway, and increased intracellular ROS (Figure 2). Repurposing the already approved marine-derived drugs should be trialed for the treatment of CRC and PC with a focus on these main mechanisms of action. This strategy offers various advantages over developing entirely new drugs [112].

In summary, MNPs have great potential for cancer treatment. However, studies of their effects on PC and CRC are scarce, and little is known about the potential pharmacokinetic interactions between MNPs and traditional tumor drugs. Additionally, clinical data about MNPs and their use in the treatment of CRC and PC are limited. The chemical diversity in the marine environment opens new horizons for various drug leads, but additional molecular targets should be explored to develop treatments for these deadly types of cancer.

## Figures and Tables

**Figure 1 ijms-23-08048-f001:**
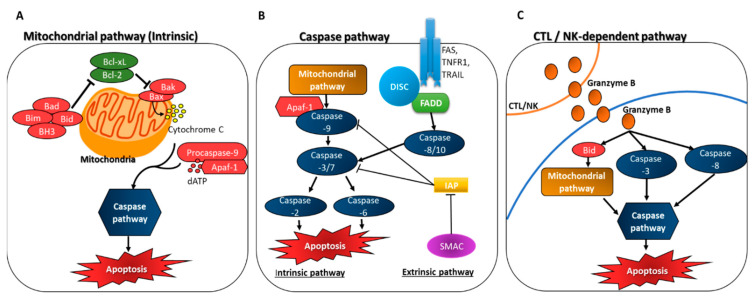
Three major pathways that induce stimuli that activate the caspase apoptosis pathway: (**A**) Mitochondrial pathway (intrinsic); (**B**) Caspase pathway; (**C**) CTL/NK dependent pathway. Bcl-extra-large (Bcl-xL); Bcl-2-associated X protein (Bax); Bcl-2-antagonist/killer (Bak); Bcl-2 associated agonist of cell death (Bad); Bcl-2 interacting mediator of cell death (Bim); Bcl-2 homology-3 (BH3); BH3-interacting domain death agonist (Bid); Apoptosis protease-activating factor-1 (Apaf-1); Deoxyadenosine triphosphate (dATP); Death-inducing signaling complex (DISC); FAS- associated death domain protein (FADD); TNF receptor 1 (TNFR1); TNF-related apoptosis-inducing ligand receptors (TRAIL); inhibitor of apoptosis protein (IAP); Second mitochondria-derived activator of caspases (SMAC); Cytotoxic T lymphocytes/natural killer cells (CTL/NK).

**Figure 2 ijms-23-08048-f002:**
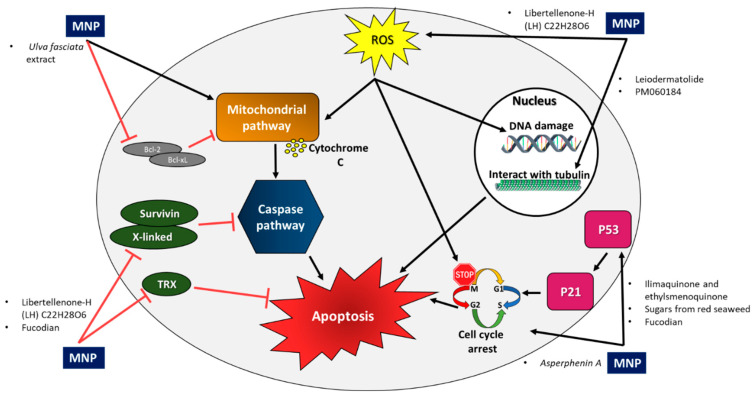
Main mechanisms of action of marine natural products (MNPs) on colon and pancreatic cancer. **Caspase**
**activation**
**pathway:** Caspases are a family of protease enzymes that play essential roles in apoptosis, and they can be activated by the **mitochondrial pathway**. The permeability of the mitochondrial outer membrane is the key mechanism for caspase activation. Release of mitochondrial components, such as cytochrome c, leads to caspase-9 activation, which leads to activation of downstream caspases (-3, -6, and -7) and induction of apoptosis. Pro-apoptotic and anti-apoptotic members of the **Bcl-2** protein family control the permeability of the outer mitochondrial membrane and regulate the release of the mitochondrial intermembrane space constituents, including cytochrome c. MNPs **interact with tubulin,** causing antimitotic activity. Microtubules are cytoskeletal elements that play an important role in many cellular functions, including cell division. **Cell cycle arrest:** The cell cycle is a four-stage process. In the G1, G2, and M stages, the integrity of DNA is assessed by cyclin-dependent kinases (CDKs). Any disruption to these stages causes cell cycle arrest, which leads to apoptosis. In addition, **p53** can induce S phase or G2/M arrest via **P21** inhibition of CDKs. The presence of reactive oxygen species (**ROS**) leads to cell cycle arrest at the G2/M phase, followed by DNA damage accumulation and induction of apoptosis. **X-linked and survivin** are members of the inhibitor of apoptosis protein family, and they prevent apoptosis by binding to caspases, including initiators and effectors, and thereby protect them from cleavage and activation. The **thioredoxin (TRX) system** is expressed in all living cells and has a variety of biological functions. It helps tumor cells evade apoptosis through binding to the apoptosis signal regulating kinase (ASK1) directly and to the tumor suppressor PTEN.

**Table 1 ijms-23-08048-t001:** Marine origin with anti-colorectal cancer activity.

Kingdom	Species	Extraction/Chemical Group	Key Findings	Cell Line/Animal Model/Epidemiology	Reference
Animalia	Soft coral: *Sinularia flexibilis*	Flexibilide C20H30O4	Regulates several metabolic pathways, including glycerophospholipid and sphingolipid, leading to activation of caspases and induction of apoptosis	HCT-116 cells	[34]
			Downregulates the tricarboxylic acid cycle (TCA), which leads to the loss of mitochondrial transmembrane potential and cell apoptosis		
			Upregulates sphingosine-1-phosphate		
	Soft coral: *Sinularia* sp.	Peroxy sesquiterpenoids	Increases production of H_2_O_2_	HCT-116 cells	[35]
			Inhibits anti-apoptosis proteins, such as Bcl-xL and pAkt, leading to apoptosis		
			Suppresses hemeoxygenase-1 (HO-1), nuclear factor-erythroid-2-related factor (Nrf2), and phosphoNrf2 (pNrf2)		
	Soft coral: *Carotalcyon* sp.	DCM/MeOH extract	Causes G2/M transition and induction of apoptosis	HGUE-C-1, HT-29, and SW-480 cells	[36]
	Nudibranchs: *Phyllidia varicosa* and Dolabella auricularia		Increase intracellular reactive oxygen species (ROS) accumulation, mitochondrial depolarization, caspase activation, and DNA damage that leads to apoptosis		
	Holothurian: *Pseudocol ochirus violaceus*		Extract causes necrotic cell death		
	Mollusc: *Dicathais orbita*	Chloroform/MeOH extract	Apoptotic response to a genotoxic carcinogen without any apparent toxic side effects	In vivo model: mice	[37]
			Contains 6-bromoisatin, which significantly enhances apoptosis and reduces cell proliferation		
	Sponge: *Lithoplocamia lithistoides*	PM060184 C_31_H_45_N_3_O_7_	Inhibits tubulin polymerization, which reduces microtubule dynamicity, and inhibits cell migration	HCT-116 cells	[38]
	Sponges: *Smenospongia aurea, S. cerebriformis*, and *Verongula rigida*	Monohydroxy-1,4-benzoquinones	Induce expression of tumor suppressor gene P53	HCT-116 and RKO cells	[39]
			Stabilize the tumor suppressor gene p53 through phosphorylation at Ser15		
		1,4-naphthoquinone	Upregulates the expression of p21WAF1/CIP1, a p53-dependent gene, and suppresses proliferation of cancer cells		
			Causes G2/M cell cycle arrest and increases caspase-3 cleavage; induces apoptosis		
	Sponge: *Hippospongia metachromia*	Ilimaquinone and ethylsmenoquinone	Activat p53 pathway and upregulate the expression of p21	HCT116 and RKO cells	[40]
			Causes G2/M cell cycle arrest		
			Increases caspase-3 cleavage		
	Fish	Fatty acids	Improve survival among stage III CRC patients with wild-type KRAS (Kirsten rat sarcoma virus)	Epidemiology research	[41]
Algae	Red seaweed	Carbohydrate	Inhibits proliferation of colon cancer cells and induces apoptosis	HCT-116 cells	[42]
			Activates caspases -3 and -9, which leads to apoptosis		
			Reduces the expression levels of anti-apoptotic proteins Bcl-2 and Bcl-xL and enhances the expression of the pro-apoptotic protein Bax		
			Induces P5, which is involved in apoptosis		
	Seaweeds: Miyeok and Dashima	Dietary intake	Lower the risk of CRC associated with the cellular Myelocytomatosis (c-MYC)	Epidemiology	[43]
			Inhibit proliferation and induce apoptosis via regulation of the WNT/β-catenin signaling pathway		
	Green algae: *Ulva fasciata*	EtOH 80% extract	Causes morphological changes indicative of apoptosis (apoptotic bodies, DNA fragmentation, and mitochondrial membrane depolarization)	HCT-116 cells	[44]
			Induces the mitochondrial intrinsic pathway by lowering Bcl-2 regulation and raising Bax regulation, subsequently upregulating caspases -9 and -3 and increasing mitochondrial membrane permeability		
	Brown algae: *Fucus vesiculosus*	Sulfated polysaccharide Fucoidan	Increases levels of cleaved caspases -8, -9, -7, and -3 and cleaved poly (ADP-ribose) polymerase, leading to induction of apoptosis in HT-29 cells	HT-29 and HCT-116 cells	[45]
			Attenuates the levels of the X-linked inhibitor of apoptosis protein and survivin		
			Enhances mitochondrial membrane permeability as well as cytochrome c and Smac/Diablo release from the mitochondria, leading to apoptosis		
			Increases the levels of the pro-apoptotic proteins Bak and truncated Bid and reduces the levels of the anti-apoptotic protein Mcl-1		
			Increases the levels of the tumor necrosis factor-related apoptosis-inducing ligand, Fas, and death receptor proteins		
			Activates the AKT pathway		[46]
	Red algae: *Asparagopsis armata* and *Sphaerococcus coronopifolius*	Crude methanol and dichloromethane extracts	Decreases Caco-2 cell proliferation	Caco-2 cells	[47]
Fungus	*Aspergillus* sp.	Ketone aryl	Inhibits the growth of cells through G2/M cell cycle arrest followed by apoptosis	RKO cells	[48]
			Triggers microtubule disassembly and induces ROS		
		Diketopiperazine disulfides + a new aranotin derivative deoxyapoaranotin	Strain KMD 901 shows potent cytotoxic activity towards five cancer cell lines (HCT-116, AGS, A549, MCF-7 and HepG2)	HCT-116 cells	[49]
			Has a direct cytotoxic and apoptosis-inducing impact towards HCT-116 cells		
Bacteria	Cyanobacteria: *cf. Neolyngbya* sp.	Lipopeptides	Signifcantly reduces cell viability by an unknown mechanism	HCT-116 cells	[50]
	Cyanobacteria: *Cyanobium* sp.	Acetonitrile fractions extracts	Affects Bcl-2 expression and alters protein networks from endoplasmic reticulum stress to proteasome degradation and apoptosis	RKO cells	[51]
	Cyanobacteria: *Synechocystis salina*		Affects progression of the cell cycle at the G2/M transition		
	Cyanobacteria: *Symploca* sp.	Peptides Dolastatins	The combination of largazole and dolastatin 10 curbs the growth of HCT-116 cancer cells	HCT-116 cells	[52,53]

**Table 2 ijms-23-08048-t002:** Marine origin with anti-pancreatic cancer activity.

Kingdom	Species	Extraction/Chemical Group	Key Findings	Cell Line/Animal Model/Epidemiology	Reference
Animalia	Sponge: *Amphibleptula*	Acetonitrile fractions extracts	Induces apoptosis in the AsPC-1, BxPC-3, and PANC-1 cell lines but not in the MIA PaCa-2 cell line	AsPC-1, BxPC-3, MIA PaCa-2, and PANC-1 cells	[54]
	Sponge: *Batzella* sp.		Inhibits NFκB transcriptional activity		
	Sponge: *Haliclona*	Alkaloid	Affects vacuolar ATPase activity and significantly increases the level of the autophagosome markers LC3-II and p62/SQSTM1	AsPC-1 and PANC-1 cells	[55]
	Sponge: *Batzella* sp.		Causes cell cycle arrest by intercalating into DNA and/or inhibiting topoisomerase II activity	AsPC-1, Panc-1, BxPC-3, and MIA PaCa2 cells	[56]
			Has low toxicity against normal cells		
	Sponge: *Leiodermatium*	Polyketide macrolide Leiodermatolide	Is a potent antimitotic agent	AsPC-1, BxPC-3 and MIA PaCa-2 cells	[57]
			Reduces cell viability and causes cell cycle arrest at the G2/M phase		
			Interacts with tubulin, causing antimitotic activity		
	Sponge: *Amphibleptula*	Methanolic extraction	induced apoptosis in the AsPC-1, BxPC-3 and PANC-1 cell lines and not to MIA PaCa-2 cell line	AsPC-1, BxPC-3, PANC-1 and MIA PaCa-2 cells	[54]
			inhibit NFκB transcriptional activity		
	Brown algae: *Ecklonia cava*	Polyphenols	Increases ROS levels	PANC-1 cells	[58]
			Induces apoptosis		
			Decreases the expression of cell progression inducers PCNA and Cyclin D1 as well as anti-apoptotic protein Bcl2 and increases the expression of the pro-apoptotic protein Bax		
			Reduces the antioxidant defense system in cancer cells without increasing inflammatory cytokine levels		
			Suppresses microtubules, appearance of multipolar mitosis, and lagging chromosomes at the metaphase plate		
Algae	Brown algae: *Turbinaria conoides*	Sulfated polysaccharide Fucoidan	Fucoidan nanoparticles loaded with quinacrine drug can reduce growth and metastasis of pancreatic cancer	In vivo model–mice and PANC-1 cells	[59]
			Induce apoptosis, activates caspases -3, -8, and -9, and cleaves Poly ADP ribose polymerase (PARP)	PANC-1 cells	[60]
		Fucoidan fraction–F5	Inhibits 57 and 38 nuclear factor κB (NFκB) pathway molecules		
			Increases cellular p53 and revert NFκB expression		
	Brown algae: *Hormophysa triquetra, Spatoglossum asperum*, and *Padina tetrastromatica*	Polyphenols	Cause death of cancer cells	Panc-1, MiaPaCa-2, Panc-3.27, and BxPC-3 cells	[39]
			Inhibit tumor growth in xenograft mice after radiation therapy	MiaPaCa-2 cell line based on xenograft mice	
Fungus	Arctic fungi: *Eutypella* sp.	Libertellenone-H (LH) C22H28O6	Increases the ROS level	PANC-1, SW1990, AsPC-1, and BxPC-3 cells	[61]
			Inhibits the thioredoxin system (TRX)		

**Table 3 ijms-23-08048-t003:** Different and common mechanisms of action of MNPs in colon and pancreatic cancer.

Colon Cancer	Pancreatic Cancer
Upregulate sphingosine-1-phosphate expression	Inhibit NFκB transcriptional activity
Increase DNA fragmentation and damage	Inhibit the JAK/STAT signaling pathway
Induce reactive oxygen species (ROS)	
Downregulate the tricarboxylic acid cycle, leading to the loss of mitochondrial transmembrane potential	
Affect the mitochondrial apoptotic pathway through depolarization and membrane permeability and release of cytochrome c and SMAC/Diablo	Increase the levels of auto-phagosome marker LC3-II and p62/SQSTM2
Increase the levels of the tumor necrosis factor-related apoptosis-inducing ligand, Fas, and death receptor 5 protein	
Suppress proteins related to the cell survival regulation signal of Nrf2-ARE (an antioxidant response element)	Affect vacuolar ATPase activity
Induce apoptosis through the caspase independent pathway	Reduce the antioxidant defense system
Activate the AKT pathway	
Cleave poly (ADP-ribose) polymerase (PARP)	Decrease the expression of tumor cell progression inducers (proliferating cell nuclear antigen and cyclin D1)
Regulate the WNT/β-catenin signaling pathway	
Alter protein networks, from endoplasmic reticulum stress to proteasome degradation, leading to induction of apoptosis	
**Common pathways**	
Induce apoptosis by caspase activation through the extrinsic and the intrinsic pathways	
Inhibit anti-apoptotic factors thioredoxin (TRX), X-linked (XIAP), and survivin	
Interact with tubulin to cause anti-mitotic activity	
Suppress cell cycle progression; cause cell cycle arrest at the G2/M phase	
Activate the p53 pathway	
Increase intracellular ROS accumulation and induction of apoptosis	

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
