# Peer review of "Natural Products of Marine Origin for the Treatment of Colorectal and Pancreatic Cancers: Mechanisms and Potential"

_ijms, 2022, doi:10.3390/ijms23148048_

Round 1

Reviewer 1 Report

The article of Nasrin Fares Amer and Tal Luzzatto Knaan deals with a topic of great interest: Natural products for the treatment of cancer (colorectal and pancreatic cancers). They focus their investigation on compounds of marine origin, a huge source o bioactive molecules. 

The article is interesting however it needs some revision:

69-70. Please, write tRNAs, ribosome, cell growth, and proliferation in normal font without linline.

In the second paragraph please briefly describe the side effect commonly found in those compounds described. 

Table 2.

please better describe in the table the compounds reported. In some boxes any compound is present. Please better format the table.

why ilimaquinone and ethylsmenoquinone or libertellenone-H (LH) (isolated from Arctic marine fungi) and some other compounds, cited in the text, are not present in the table? Leiodermatolide, PM060184 Other examples. 

Figure 2 would be better if the compounds with their name were integrated within the action mechanisms of apoptosis

Reviewer 2 Report

The review article by Amer and Knaan summarizes the information on the natural products derived from marine sources for which the anticancer activity in colorectal and pancreatic cancers was reported. While the main part of the article reporting mechanisms of action is well-written and comprehensive, where are two main issues which need to be met before the manuscript is acceptable for publication.

1)      The information on marine-derived drugs currently approved for clinical use must be updated. In the recent years there were some new drugs approved for the use in clinic, e.g. Aidixi™, PADCEV™ and TIVDAK™ should be mentioned. Please refer to the https://doi.org/10.1096/fasebj.2022.36.S1.L7586, https://doi.org/10.1096/fasebj.2020.34.s1.01808, https://doi.org/10.3390/md18120643.

2)      One of the strategies to decrease the cancer-related death rate, which is especially important in colorectal and pancreatic, is cancer prevention by modification of a diet. Higher consumption of sea food is known to be a promising strategy to prevent both cancer types. Therefore, the chapter summarizing cancer-preventive properties of marine-derived compounds is to be included in the current review to make it more comprehensive. Here please refer to https://doi.org/10.3390/md12020636 and https://doi.org/10.3390/md19100558.

Round 2

Reviewer 1 Report

Table 1.  please better describe in the table the compounds reported. In some boxes any compound is present in the column of Chemical group. Please better format the table.

Author Response

Dear Editor,

Following Reviewer’s 1 comment, we have added as much information we could gather regarding the chemical group. As many papers did not report a single compound and many are extracts, we have added the extraction information as well. We have split Table1 to two tables (1and 2), and the previous table 2 is now Table 3 (we have changed the referral in the text accordingly). In terms of formatting, we changed the table layout to landscape (attached file) and if your editorial team can add this to the main text, that would be most helpful.

We can only upload a single file here, therefore we uploaded the table, and will send the main text and tables TC format via email.

We appreciate the comment and the help for making our manuscript better. We hope you will find it worthy of publication in your journal.

Sincerely,

Tal
